# Regional Variation of Chemical Characteristics in Young Marselan (*Vitis vinifera* L.) Red Wines from Five Regions of China

**DOI:** 10.3390/foods11060787

**Published:** 2022-03-09

**Authors:** Yibin Lan, Min Liu, Xinke Zhang, Siyu Li, Ying Shi, Changqing Duan

**Affiliations:** 1Center for Viticulture & Enology, College of Food Science and Nutritional Engineering, China Agricultural University, Beijing 100083, China; lanyibin@cau.edu.cn (Y.L.); liu_min21@163.com (M.L.); zhangxinke@bua.edu.cn (X.Z.); siyuli@cau.edu.cn (S.L.); shiy@cau.edu.cn (Y.S.); 2Key Laboratory of Viticulture and Enology, Ministry of Agriculture and Rural Affairs, Beijing 100083, China; 3Food Science and Engineering College, Beijing University of Agriculture, Beijing 102206, China

**Keywords:** Marselan, geographical origin, GC–MS, HPLC–QqQ–MS/MS, sensory

## Abstract

The environmental conditions of wine regions determine the flavor characteristics of wine. The characterization of the chemical composition and sensory profiles of young Marselan wines from five wine-producing regions in China was investigated by gas chromatography-mass spectrometry (MS), high-performance liquid chromatography–triple-quadrupole MS/MS and descriptive analysis. Young Marselan wines can be successful discriminated based on concentrations of volatile compounds, but not phenolic compounds, by orthogonal partial least squares discriminant analysis according to regions. Compared to Jiaodong Peninsula (JDP) and Bohai Bay (BHB) regions, there were relatively lower average concentrations of varietal volatiles (mainly including *β*-citronellol, geraniol, and (*E*)-*β*-damasenone) and several fermentation aroma compounds (including isoamyl acetate, octanoic acid, decanoic acid, ethyl decanoate, etc.) but higher levels of acetic acid in Xinjiang (XJ), Loess Plateau (LP), and Huaizhuo Basin (HZB) regions, which were related to their characteristic environmental conditions. Marselan wines from HZB, LP, and XJ regions were characterized by lower *L* values and higher *a* and *C*_ab_ values. Marselan wines from XJ were discriminated from the wines from other regions due to their higher concentrations of several flavonols. Sensory analysis indicated that Marselan wines from HZB region were characterized by relatively low intensities of floral and fruity aromas compared to other regions.

## 1. Introduction

Marselan (*Vitis vinifera* L.) is a particularly successful cross of Cabernet Sauvignon and Grenache Noir obtained by Institut national de la recherche agronomique (INRA) in 1961 (Appendix A). This variety shows good resistance to powdery mildew, mites, and especially to botrytis bunch rot [1]. Marselan was first introduced into China in 2001 and was planted in the Huaizhuo Basin region, Hebei Province. Due to its strong disease resistance and small berries with high quality, Marselan has been popular and planted almost wherever vines are grown for wine production in China, except for cool wine-producing regions. Followed by France, China has the second-largest planting area of Marselan [2], and the planting area is still growing.

During the past decade, several researchers have focused on the phenolic compounds in Marselan grapes, such as the phenolic compounds in skins [3] and changes in the different degrees of polymerization in seeds [4]. In addition, effects of some viticultural practices on the chemical compositions in Marselan grape berries were studied, such as grape cluster shading [5] and flower cluster tip removal [6], and influences of several enological parameters on Marselan wines were investigated, such as application of non-*Saccharomyces* [7] and spontaneous fermentation [8]. Wines made from the Marselan variety have aromas of black fruit, spices, cocoa, and vegetal characteristics [9]. More recently, key odorants of Marselan dry wine from China were identified by the systematic ‘sensomics’ approach, which includes gas chromatography–olfactometry (GC-O), quantitative measurements, aroma recombination, and omission tests [10]. The aroma profile of Marselan wine has been described as blackberry, green pepper, honey, raspberry, caramel, smoke, and cinnamon [10]. In addition, young Marselan wine in China is characterized by intense of floral aromas due to its relatively high concentration of terpenes [11].

Wine aroma and taste are major factors that determine the characters and quality of wine, and are influenced by grape variety, geographical region, viticultural practice, and vinification technique. Generally, a specific variety can be qualitatively or quantitatively characterized by the varietal aroma compounds and phenolic compounds, determining its distinctive sensory characteristics, such as color, aroma, and taste. Geographical origin (GI) is considered a synonym for both wine style and quality by producers and consumers. The environmental conditions (e.g., soil, sunlight, heat, water, aspect, and altitude) of geographical origins have a vital influence on the chemical compositions (especially the volatile aroma compounds and phenolic compounds) of final wines [12,13]. Consequently, monovarietal wines from different regions have been produced with different flavor characteristics. Therefore, consumers can recognize the geographical origin of a specific wine through sensory tasting, and many researchers have successfully classified wines according to geographical origin based on quantitative results of the aroma and phenolic compounds [14].

In this study, the main aims were to investigate the chromatographic fingerprints of Marselan wines from different wine-producing regions in China and identify the flavor compound characteristics in different regions. In addition, this study also investigated and discusses the differences in environmental conditions between five wine-producing regions that drive regional differences in Marselan wines.

## 2. Materials and Methods

### 2.1. Reagents and Standards

Analytical grade chemicals, including NaCl, NaOH, tartaric acid, citric acid, glucose, disodium hydrogen phosphate, and anhydrous sodium sulfate, were supplied by Beijing Chemical Works (Beijing, China). The chromatographic grade solvents, including methanol, ethanol, dichloromethane, acetonitrile, and formic acid, were purchased from Honeywell Burdick & Jackson (Morristown, NJ, USA). Ultrapure water was obtained from a Milli-Q Element water purification system (Millipore, Milford, MA, USA). The aroma standards and C_6_-C_24_ *n*-alkanes were obtained from Sigma-Aldrich (St. Louis, MO, USA). The anthocyanin and non-anthocyanin standard compounds were purchased from Sigma-Aldrich (St. Louis, MO, USA), ChromaDex (Irvine, CA, USA), and Extrasynthese (Genay, France).

### 2.2. Wine Samples

A total of 38 commercial Marselan red wines from five important wine-producing regions, namely Jiaodong Peninsula (JP), Bohai Bay (BHB), Huaizhuo Basin (HZB), Loess Plateau (LP), and Xinjiang (XJ), were used in this study (Appendix A). The wines ranged in vintage from 2012 to 2016, and their accurate subregions are presented in Appendix A and plotted on a China map (Appendix A). All were produced as premium wines, which go for a period of bottle aging. The basic wine compositions (including alcohol, reducing sugar, total acidity, pH, and glycerol) of the Marselan wines were measured by a WineScan (FT 120) rapid-scanning infrared Fourier-transform spectrometer with FOSS WineScan software version 2.2.1 (Foss Electric, Hillerød, Denmark). Wines samples were centrifuged at 4000× *g* for 8 min and the analyzed following the manufacturer’s instructions. Colorimetric measurement, analysis of volatile compounds and phenolic compounds, and descriptive analysis were carried out on these wines and completed at the end of 2017.

### 2.3. Meteorological Data

The meteorological data (except for the photosynthetically active radiation (PAR)) of these five wine regions, from 2000 to 2017 vintages, were obtained from the National Meteorological Information center [15]. The PAR data of these wine regions, from 2000 to 2014 vintages, were obtained from an open access dataset, named “A dataset of reconstructed photosynthetically active radiation (PAR) in China (1961–2014)” [16]. In this study, the sunshine duration, PAR, precipitation, average daily temperature, and diurnal temperature range in the growing season (from May to October) for each wine region were calculated.

### 2.4. Colorimetric Measurement

Colorimetric measurement was carried out on a UV–visible spectrophotometer (Shimadzu UV-2450, Shimadzu Co., Kyoto, Japan), recording the wine absorbance spectra (380–700 nm). Ultra-pure water was used as blank. The values of the CIELab parameters, including chromaticity (*Cab*), lightness (*L*), hue (*Hab*), red-greenness (*a*), and yellow-blueness (*b*), were calculated according to a previous reported method [17]. Prior to analyses, all wine samples were filtered through 0.45 μm cellulose filters (Membrana Co., Wuppertal, Germany) and placed in a 2 mm path length quartz cuvette. All the experiments were performed in triplicate.

### 2.5. HS–SPME–GC–MS Analysis of Aroma Compounds

The extraction and identification of the volatile compounds were carried out according to the analytical method previously reported in our laboratory, with slight modifications [18]. The analysis of the volatile compounds in wines was performed on an Agilent 6890 chromatograph (GC) coupled to an Agilent 5975C mass spectrometer (MS, Agilent Technologies, Santa Clara, CA, USA). A CTC CombiPAL autosampler (CTC Analytics AG, Zwingen, Switzerland) with a 50/30 μm divinylbenzene/carboxen/polydimethylsiloxane (DVB/CAR/PDMS) fiber (2 cm, Supelco, Bellefonte, PA, USA) was used for the extraction of the volatile compounds. A wine sample of 5 mL was transferred into a 20 mL vial with 10 μL of 4-methyl-2-pentanol (internal standard, 1.0086 g/L) and 1 g NaCl. Tightly capped with a PTFE-silicon septum, the samples were equilibrated at 40 °C for 30 min before extraction by an SPME fiber at 40 °C for 30 min with stirring at 500 rpm. After extraction, the fiber was thermally desorbed by insertion into the injection port of the GC for 8 min. The injection was performed in splitless mode (0.75 min) with a GC inlet temperature of 250 °C.

The separation of the volatile compounds was carried out on a HP-INNOWAX capillary column (60 m × 0.25 mm × 0.25 μm, J&W Scientific, Folsom, CA, USA). The flow rate of the carrier gas (helium) was 1 mL/min. The oven temperature was held at 50 °C for 1 min after injection, then programmed to 220 °C at a rate of 3 °C/min, and kept at 220 °C for 5 min. The temperatures of the mass selective detector transfer line, ion source, and quadrupole were 250 °C, 230 °C, and 150 °C, respectively. The positive ion electron impact spectra at 70 eV were recorded with a mass range of 30–350 *m*/*z* for the full scan mode.

The quantification process was carried out according to our previous study [18]. A synthetic model wine solution was prepared in distilled water containing 14% *v*/*v* ethanol, 5 g/L tartaric acid, and the pH was adjusted to 3.3 with a 5 M NaOH solution. The calibration curves of each aroma standard were obtained according to fifteen dilution levels in succession with the model wine solution.

### 2.6. HPLC–QqQ–MS/MS Analysis of Phenolic Compounds

The analyses of the phenolic compounds were performed on an Agilent 1200 series high-performance liquid chromatographer (HPLC) equipped with an Agilent 6410B triple-quadrupole (QqQ) mass spectrometer (MS) (Agilent Technologies, CA, USA), based on our previous studies [19,20]. The chromatographic separation was carried out using a Poroshell 120 EC-C18 column (150 mm × 2.1 mm, 2.7 μm; Agilent Technologies, Santa Clara, CA, USA). The mobile phases were: A = water solution containing 0.1% (*v*/*v*) formic acid; B = acetonitrile/methanol solution (50:50, *v*/*v*) containing 0.1% (*v*/*v*) formic acid. The gradient elution was: (1) from 10% to 46% B in 28 min, and (2) from 46% to 10% B in 1 min. The post time was 5 min. A wine sample of 1 mL was filtered using 0.45 μm inorganic membranes (polyether sulphone). The injection volume was 5 μL and the flow rate was 0.4 mL/min. The column was thermostatically controlled at 55 °C. An electrospray ionization source was used with 4 kV voltage in the negative mode and the positive mode for the non-anthocyanin phenolics and the anthocyanin compounds, respectively. The temperatures of the ion source and gas (N_2_) were 150 °C and 350 °C, respectively. The gas had a flow rate of 12 L/min, while the nebulizer pressure was 35 psi. The [M + H]+ and [M + H]− ions were selected as the precursors for the anthocyanin and non-anthocyanin compounds, respectively. The multiple reaction monitoring mode was used for both identification and quantification. The identification was achieved by comparing the retention times and qualitative transition ions with the phenolic compounds in an in-home phenolic compound library [19,20]. The quantification process was carried out according to reliable methods established in our previous studies [19,20]. All anthocyanins were quantified on the basis of the calibration curve of malvidin-3-*O*-glucoside and non-anthocyanins were quantified according to the calibration curves of their own reference compounds.

### 2.7. Descriptive Analysis

Descriptive analysis was conducted in a sensory laboratory equipped with 20 individual booths at controlled room temperature (20 °C). Approximately 30 mL of each of the wines was prepared in International Standards Organization (ISO) wine tasting glasses (ISO 3591:1977). The glasses were labeled with three-digit random numbers and served in a randomized order.

A panel consisting of 16 assessors (8 males and 8 females) was convened for sensory evaluation of Marselan wines from different wine-producing regions. All panelists were student or staff or faculty at the Center for Viticulture and Enology (CFVE), China Agricultural University. All panelists attended four one-hour training sessions. In the first session, all panelists were asked to evaluate all Marselan wines used in this study and generate descriptive terms, including 21 aroma and taste attributes. In the second session, panelists were asked to discuss the terms and seven attributes (hue, color intensity, floral, overall fruity, herbaceous, acidity, and astringency) were selected which could adequately characterize the sensory difference among targeted wines. In the third session, all reference standards representative of attributes were prepared to train the panelists. In the fourth session, all panelists were asked to recognize the attributes and rate the intensities of reference standards. Panel performance was assessed after training, including discrimination ability, repeatability, and reproducibility, according to the method in our previous study [21].

For formal session, panelists were asked to rate the intensity of the attributes of wine samples from 1 (weak) to 10 (strong) using a 10-point scale according to a previous study [22].

### 2.8. Statistical Analysis

A one-way analysis of variance (ANOVA) was conducted to determine any significant differences between wines of different regions by using the aov and Duncan.test functions in the agricolae package of the R software environment (version 3.0.3, http://www.r-project.org/, accessed on 10 October 2021). A significance level α = 0.05 was used for the statistical evaluation. Orthogonal partial least squares discriminant analysis (OPLS-DA) was performed using soft independent modeling of class analogy (SIMCA, version 14.1 from Umetrics), to classify wine samples into different classes based on wine-producing region and identify the chemical variables most responsible for the differentiations. All the chemical variables were normalized before the multivariate statistical analysis. Boxplots for the environmental factors were prepared with the ggplot2 package in the R environment.

## 3. Results and Discussion

### 3.1. Meteorological Data

The climatic conditions of wine regions in China vary considerably, leading to significant variation in the quality and styles of wine. According to the Köppen-Geiger climate classification system, which is a widely used vegetation-based empirical climate classification system [23], the JDP region has a monsoon-influenced humid subtropical climate, and BHB has a monsoon-influenced hot-summer humid continental climate (Appendix A). Meanwhile, the HZB, LP, and XJ regions have either a cold semi-arid climate or cold desert climate. Further investigation of the climatic conditions of wine-producing regions was achieved by analyzing the meteorological data. The climate results showed that the XJ region is characterized by the longest sunshine duration (on average, 9.7 h), highest PAR (on average, 32.98 mol/(m^2^.d) in Yanqi County), and diurnal temperature range (on average, 13.07 °C in Manasi County and 15.10 °C in Yanqi County), with the lowest rainfall (on average, 115.88 mm in Manasi County and 61.83 mm in Yanqi County) (Figure 1). In contrast, the JDP and BHB regions displayed significantly shorter sunshine durations, lower PARs, and diurnal temperature ranges, with significantly higher rainfall. These regions have either a warm or hot climate. The Linfen and Qingdao subregions, of LP and JDP, respectively, have a relatively higher average daily temperature than the other regions due to their relatively lower latitudes (Figure 1 and Appendix A). In addition, a comprehensive investigation of the climatic characteristics of wine-producing regions in China was carried out according to a newly established zoning index system named the *FRD-DI-T* system [24]. It found that the HZB, LP, and XJ regions (*DI* > 1.6) showed higher levels of the dryness index (*DI*) when compared with those of BHB and JDP (*DI* ≤ 1.6), even though HZB is near to the BHB and JDP regions.

### 3.2. Basic Wine Compositions

The ANOVA results of the basic wine compositions, including ethanol, reducing sugar, total acidity, pH, and glycerol, are presented in Table 1. The Marselan wines from XJ, LP, and HZB demonstrated a higher mean concentration of ethanol than the wines from JDP and BHB. The high diurnal temperature ranges in XJ, LP, and HZB, as well as the application of the late harvest technique in HZB, could explain these phenomena. Reducing sugars showed higher concentrations in wines from XJ, followed by wines from LP. Similar to alcohol results, glycerol in wines from XJ, LP, and HZB displayed relatively a higher mean concentration compared with wines from JDP and BHB. No differences existed between the pH levels. Excluding JDP, similar pH results were found in the total acidity of the other four regions. This could be because the vinification technique of adjusting acidity with tartaric acid is widely applied in the wine production in western regions (e.g., XJ, LP, and Ningxia).

### 3.3. Aroma Compounds

The OPLS-DA was applied to identify the most characteristic marker compounds for wine region discrimination. As shown in Figure 2A,B, a clear separation was obtained by a reliable OPLS-DA model (*R*^2^*X* = 0.708, *R*^2^*Y* = 0.799, and *Q*^2^ = 0.545), except for the similarity between the XJ and LP Marselan wines. Interestingly, the OPLS-DA models showed a clear separation between two classes (Class 1 includes JDP and BHB; Class 2 includes HZB, LP, and XJ) based on the first component. It is well known that the discrimination of similar-style wines made from the same variety in different regions is typically ascribed to differences in the environmental conditions.

The volatile aroma compounds that showed differences (*p* < 0.05) across the regions are presented in Table 2. As expected, the majority of the potential maker compounds for discrimination displayed differences in concentration. In order to interpret the volatile aroma compound differences between varieties, the potential marker compounds with differences are summarized and discussed according to their origins (e.g., grapes and fermentation) and chemical categories (e.g., terpenes, norisoprenoids, ethyl esters, and acetates).

#### 3.3.1. Terpenes

As shown in Table 2, four terpenes were identified in Marselan wines, including linalool, 4-terpineol, *β*-citronellol, and geraniol, which was similar to results of a previous study [10]. The concentration of monoterpenol was higher compared with neutral grape varieties, such as Cabernet Sauvignon [11]. Interestingly, the concentration of geraniol in wines from BHB region exceeded its odor threshold (30 μg/L, detected in 10% ethanol/water (*v*/*v*) solution) [25], which might directly contribute to the wines’ floral note.

For terpenes, there were higher average concentrations of *β*-citronellol and geraniol in JDP and BHB regions compared to those in the HZB, LP, and XJ regions, except for geraniol in HZB, which showed no difference between JDP and BHB (Table 2). These grape-derived aroma compounds could have originated directly from the grape berry or from the release of their glycoside precursors during the fermentation and storage of wine [39]. Furthermore, factors such as cultivar, climatic condition, grape maturity, and winemaking procedure could play a vital role in modulating the volatile terpene concentrations in grapes and wines, finally determining the expression of aroma characteristics [40]. Research focusing on the comparison of the aroma compounds in Cabernet Sauvignon and Merlot wines from four regions in China showed that both varieties from HBCL (a subregion of BHB, plotted in the map shown in Appendix A) contained higher concentrations of citronellol than those in HBSC (a synonym of the HZB region, Appendix A) [41]. Additionally, according to the results of our previous studies aimed at investigating the differences in the volatile compounds in Muscat Blanc à Petits Grains grapes between two regions, we found that mature grape berries from CL contained higher concentrations of free and glycosidically bound terpenes than berries from the GT region, which is considered to have a cool continental climate due to its high altitude and is also characterized by high sunshine duration, PAR, and diurnal temperature ranges, with low rainfall [42]. Previous studies concluded that the grape berries exposed to sunlight had increased content of both free and glycosidically bound monoterpenes (e.g., linalool, *β*-citronellol, and geraniol) in Gewürztraminer [43], Riesling [39], and Pinot Noir [44]. However, in this study, conflicting results were found in the comparison of Marselan wines among regions with different climates. The HZB, LP, and XJ regions have relatively high sunshine duration and PAR compared to the JDP and BHB regions. As we know, the environmental factors that affect the accumulation of the aroma compounds in grapes or wines were obtained from many viticultural trials, such as leaf removal, shading, and canopy management, which influence the microclimate conditions of the fruit zone. Until now, the difference of the terpenoid metabolism in grapes under different climatic conditions, especially monsoon-influenced climates in China, has remained largely unstudied.

#### 3.3.2. Norisoprenoids

Regarding norisoprenoids, (*E*)-*β*-damasenone had the highest OAVs compared to other volatiles, ranging from 189.4 (in XJ) to 306.2 (in JDP), which could be attributed to its relatively low odor threshold (0.05 μg/L in 10% *w*/*w* water/ethanol solution [22]) (Table 2). This result was similar to that of a previous study [10], which indicated this compound could be considered the key odor-active compound in Marselan wines. (*E*)-*β*-damasenone in JDP and BHB regions presented higher concentrations compared to those in the HZB, LP, and XJ regions (Table 2). Marais et al. (1992) reported that Riesling wines from warm or hot climates (e.g., South Africa) displayed higher concentrations of norisoprenoids (e.g., TDN and vitispirane) compared to those from cool climates (e.g., Germany and Northern Italy) [45]. Previous research in our lab was carried out on Cabernet Sauvignon grapes to investigate the differentiation in the biosynthesis of norisoprenoids between the CL and GT regions by application of the metabolomic and transcriptomics techniques [46]. However, higher concentrations of (*E*)-*β*-damascenone, as well as total norisoprenoids in mature grapes, were shown in GT (cool climate) than in CL (warm climate) [46]. According to the meteorological data, the XJ region shows similarities to the GT region in terms of the high sunshine duration, PAR, and diurnal temperature range; however, the XJ region has a significantly warmer climate during the growing season than the GT region. It seemed that a complex mechanism was involved in the effect of the environmental factors on the metabolism of norisoprenoids in grapes. Additionally, numerous researchers have investigated the influences of microclimatic conditions on the concentrations of volatile aroma compounds and their precursors. In an investigation on the effects of vine microclimate on norisoprenoid concentrations, Cabernet Sauvignon grapes without leaf removal, as well as their corresponding wines, were found to have the highest concentration of (*E*)-*β*-damascenone [47]. However, the concentrations of (*E*)-*β*-damascenone in grapes were linearly and positively correlated with increasing sunlight exposure when leaves were removed. Similar results were obtained by a leaf removal treatment performed on Pinot Noir [44]. Meanwhile, Feng et al. (2015) only found higher concentrations of (*E*)-*β*-damascenone in grapes with 100% leaf removal. Water deficiency also favors the production of bound (*E*)-*β*-damascenone in grape berries, such as in Cabernet Sauvignon [48] and Merlot [49]. It seemed that the findings from the viticultural trials related to the modulation of vineyard microclimate could not be directly applied to interpret the differences in the aroma characteristics of wines from different regions with various climatic conditions. Therefore, intensive studies should be carried out to investigate the metabolic mechanism of aroma compounds in grapes in response to either a different climate style or changes to vine microclimate by viticultural practice.

#### 3.3.3. Fermentation Aroma Compounds

Acetic acid, one of the key fermentation aroma compounds, showed higher concentrations in XJ wines compared to those from other regions, excluding wines from LP. As we know, the production of ethanol from reduced sugars during alcoholic fermentation occurs along with the production of acetic acid, as well as glycerol. The higher initial content of sugar must increase the production of acetic acid as well as glycerol by yeast metabolism [50,51], even if the initial sugar ranges from 224–268 g/L, which is much lower than that in the musts for ice wine or botrytized wine production. Yeast increases its intracellular ambulation of glycerol to counterbalance the osmotic pressure induced by high sugar content. In this study, the strongly positive correlation between glycerol and ethanol (*r* = 0.79, *n* = 39, *p* < 0.01), as well as between acetic acid and ethanol (*r* = 0.69, *p* < 0.01), reveals that the differences in the musts from different regions may be involved in regulating the yeast metabolism during alcoholic fermentation (Figure 3). Furthermore, higher initial must sugar concentrations (e.g., ice wine and botrytized wine musts) contribute to the hyperosmotic stress environment, which could upregulate the expression of *ALD3* and *GPD1* in yeast during alcoholic fermentation and lead to relatively high concentrations of acetic acid in final wines [52]. Combining the results mentioned above, relatively lower concentrations of several fermentation aroma compounds (e.g., higher alcohol acetates, ethyl esters of fatty acids, higher alcohols, fatty acids, and other esters) in wines with higher alcohol concentrations reveal that the production of volatile aroma compounds in yeast could be regulated by the initial sugar concentrations in musts. Therefore, the difference of the initial sugars in musts between cool and warm regions may play a vital role in influencing the sensory differences in final wines by modulating the yeast metabolism.

For other fermentation aroma compounds, the Marselan wines from JDP and BHB contained relatively higher concentrations of isoamyl acetate, hexyl acetate, phenethyl acetate, octanoic acid, decanoic acid, ethyl decanoate, ethyl dodecanoate, isoamyl octanoate, 1-decanol, and 1-dodecanol (Table 2). Among these differentiated aroma compounds, the OAVs of isoamyl acetate (ranging from 21.1 in HZB to 55.0 in JDP), octanoic acid (ranging from 3.6 in HZB to 4.1 in JDP), decanoic acid (ranging from 2.3 in LP to 2.5 in JDP), and ethyl decanoate (ranging from 3.5 in LP to 7.4 in JDP) were greater than 1. These results indicated that Marselan wines from JDP and BHB regions may be characterized by a higher-intensity fresh fruity aroma. These volatile compounds are formed by the yeast metabolism from various precursors (mainly dominated by sugars, amino acids, and fatty acids). Similar to acetic acid and glycerol, the differences of the fermentation aroma compounds in wines among regions could also be affected by the differences of initial sugar in musts through yeast alcoholic fermentation. Furthermore, many studies comparing the naturally sweet wines made from dehydrated grapes with the corresponding base wines made from fresh grapes have concluded that the former contained higher concentrations of fatty acids, ethyl esters of fatty acids, and higher alcohol acetates [21,53,54]. Certainly, the formation mechanism of the volatile aroma compounds during alcoholic fermentation is complex. The initial concentrations of amino acids, fatty acids, and phenolic compounds in must could also affect the yeast metabolism during alcoholic fermentation [55]. However, how the matrix of grape must affects the volatile compounds of finished wine needs to be further studied.

### 3.4. Chromatic Parameters and Phenolic Compounds

According to colorimetric measurements, Marselan wines from HZB, LP, and XJ regions displayed lower *L* values and higher values of *a* and *C*_ab_ compared with the wines from JDP and BHB (Table 3). These differences may contribute to the relatively deeper color intensity and higher red-hue intensity in wines from HZB, LP, and XJ regions. No significant differences in *b* and *H*_ab_ values existed among different wines.

The OPLS-DA model (*R*^2^*X* = 0.591, *R*^2^*Y* = 0.476, *Q*^2^ = 0.183) could not discriminate well between the Marselan wines from different regions based on the quantitative data of the phenolic compounds (Figure 2C,D). However, the Marselan wines from XJ were separated from the wines from other regions due to their higher concentrations of several flavonols (e.g., myricetin-3-*O*-glucoside, quercetin-3-*O*-galactoside, isorhamnetin-3-*O*-glucoside, syringetin-3-*O*-glucoside) (*p* < 0.05) (Table 4). This suggests that aromatic and phenolic compounds in wines from different regions were influenced by climatic conditions in a different manner. The average total flavonol concentrations in XJ Marselan wines were approximately 2.98–4.31-fold higher than the wines from other regions. This is probably because the XJ region has the longest sunshine duration and highest PAR (Figure 1), which induces a more active synthesis of flavonols to resist the UV radiation [56]. Our previous study also found that more flavonols were accumulated in grapes from the GT region before the berry harvest than in those from the CL region [57].

Lower concentrations of hydroxybenzoic acids (e.g., gallic acid) in JDP and BHB wines were observed (Table 4). Our previous studies reported that Cabernet Sauvignon wines from western regions can be distinguished from those from eastern regions in China based on the quantitative results of the phenolic compounds [19,58]. Li et al. (2017) reported that the five flavanols (including catechin, epicatechin, procyanidin B1, procyanidin B2, and procyanidin C1) and gallic acid were identified as the key phenolic compounds for regional differentiation and contained higher concentrations in western regions [19]. In this study, these five flavanols had VIP scores ≥ 1 (data not shown) and displayed the highest concentration in the Marselan wines from the XJ region than from the other regions, except for epicatechin (Table 4). However, little is known that allows us to interpret the differences in aromatic and phenolic profiles of wines from different regions with various climatic conditions.

### 3.5. Sensory Profiles

In this study, sensory analysis of young Marselan wines from five wine-producing regions was studied. Panelists evaluated the significant differences among wines in the attributes of hue, color intensity, floral, fruity, herbaceous, acidity, and astringency, and the mean intensities of those attributes and standard deviations are summarized in Table 5. The results showed that young Marselan wines from LP and XJ regions had higher color intensity than wines from other three regions, with a decreasing order of LP > XJ > BHB > HZB > JP. This result was similar to that of the *L* value described above, except that the wines from HZB had relatively low *L* values, but with light color intensity. Floral and fruity attributes were scored lowest in wines from HZB region compared to wines from other regions in this study, which was different from the results of terpenes and esters described above. Wines from XJ region had lowest acidity compared to wines from other regions, although XJ Marselan wines had relatively high levels of total acidity. This could be explained by the higher concentrations of reducing sugar and glycerol in XJ Marselan wines, which can mask the expression of organic acids. JP wines had the highest acidity, but showed no significant difference from other three regions. There were no significant differences in hue, herbaceous, and astringency attributes among five region Marselan wines in this study, which could be due to the young Marselan wines used in this study.

## 4. Conclusions

Using OPLS-DA on the volatile aroma compound data, successful differentiation of young Marselan wines according to geographical origin was achieved. Higher concentrations of *β*-citronellol, geraniol, (*E*)-*β*-damascenone, and several fermentation aroma compounds were observed in young Marselan wines from JDP and BHB regions with similar climatic conditions in grape growing season (short sunshine duration, low PARs and diurnal temperature ranges, and high rainfall). Different initial sugar concentrations in musts from different regions could influence the aroma compounds in final wines by involvement in the modulation of yeast metabolism during alcoholic fermentation. These results provide a clue for further research that aims to interpret the differences in flavor compounds between different regions. However, it is difficult to differentiate the young Marselan wines between different regions based on the phenolic compound data. This research was limited by the number of Marselan wines due to the current development situation of this variety in China. For further studies, a larger dataset of quantitative results of Marselan wines obtained by high-resolution MS would be helpful to identify the aroma and flavor characteristics from different geographical origins.

## Figures and Tables

**Figure 1 foods-11-00787-f001:**
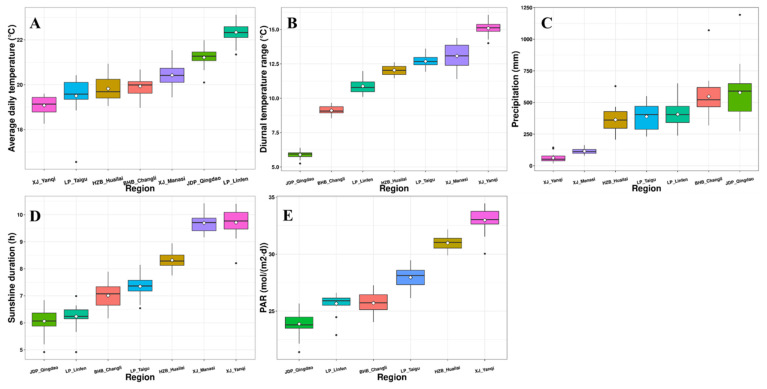
Climatic conditions of wine-producing subregions in China in this study. (**A**) Average daily temperature; (**B**) diurnal temperature range; (**C**) precipitation; (**D**) sunshine duration; (**E**) PAR. The climatic conditions of the Manasi region were based on Hutubi County. The PAR of the Taigu region was based on Taiyuan City.

**Figure 2 foods-11-00787-f002:**
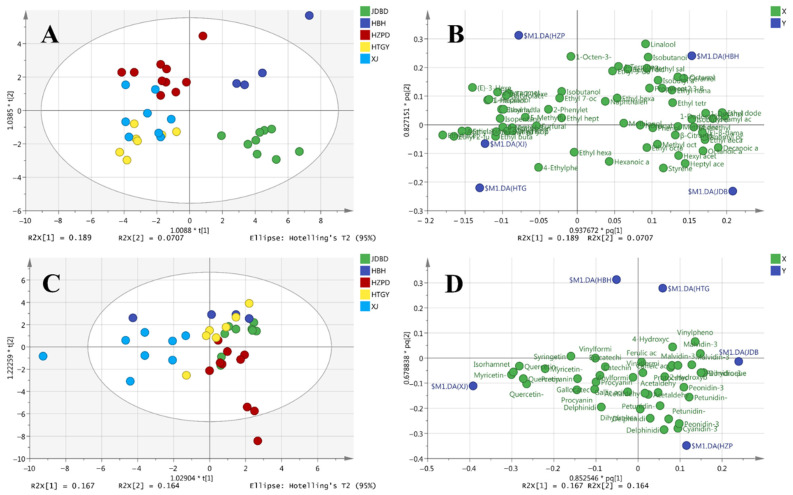
OPLS-DA models based on the flavor compounds of Marselan wine samples from different regions of China. (**A**) Score scatter plot for Marselan wine samples based on the concentrations of volatile compounds. (**B**) Loading plot for the volatile compounds of Marselan wine samples. (**C**) Score plot for Marselan wine samples based on the concentrations of phenolic compounds. (**D**) Loading plot for the phenolic compounds of Marselan wine samples.

**Figure 3 foods-11-00787-f003:**
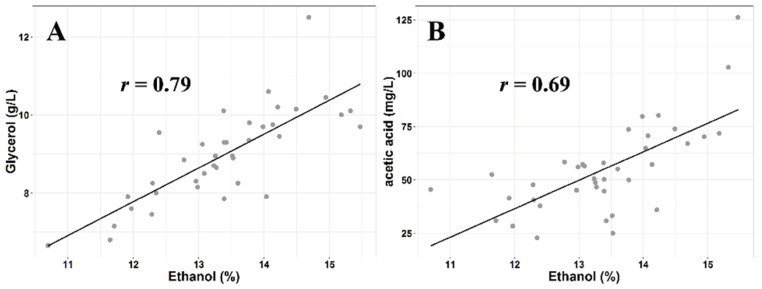
Correlation analysis between ethanol and glycerol (**A**) and acetic acid (**B**).

**Table 1 foods-11-00787-t001:** Basic parameters of Marselan wines from different regions of China *.

Region	Alcohol (%)	Reducing Sugar (g/L)	Total Acidity (g/L)	pH	Glycerol (g/L)
Jiaodong Peninsula (JDP)	12.75 ± 0.70 b	2.93 ± 0.81 c	4.87 ± 0.53 b	3.69 ± 0.07 a	8.31 ± 0.57 bc
Bohai Bay (BHB)	12.45 ± 0.79 b	3.46 ± 1.04 bc	5.18 ± 0.75 ab	3.85 ± 0.09 a	7.78 ± 0.85 c
Huaizhuo Basin (HZB)	13.54 ± 0.80 ab	3.46 ± 0.50 bc	5.84 ± 0.80 a	3.63 ± 0.18 a	9.57 ± 1.21 a
Loess Plateau (LP)	13.52 ± 1.53 ab	4.11 ± 0.41 ab	5.71 ± 1.17 ab	3.74 ± 0.32 a	9.07 ± 1.43 ab
Xinjiang (XJ)	14.24 ± 0.77 a	4.36 ± 0.92 a	5.70 ± 0.35 ab	3.77 ± 0.21 a	9.49 ± 0.94 a

* Averages and standard deviation followed by different letters are significant at *p* < 0.05, Duncan’s multiple range test.

**Table 2 foods-11-00787-t002:** Concentrations and thresholds (μg/L) of the volatile compounds of Marselan wines from different regions of China *.

Compound	Threshold ^†^	Jiaodong Peninsula	Bohai Bay	Huaizhuo Basin	Loess Plateau	Xinjiang
Ethyl acetate	7500 [25]	53,131.62 ± 12,669.67 c	70,487.82 ± 24,192.58 bc	88,681.25 ± 27,710.30 ab	120,690.88 ± 39,741.90 a	116,549.20 ± 30,546.87 a
Isobutyl acetate	1600 [26]	9.48 ± 6.33 ab	18.86 ± 13.76 a	9.21 ± 13.26 ab	7.24 ± 5.06 b	6.67 ± 4.70 b
Isoamyl acetate	30 [25]	1648.58 ± 1016.73 a	2050.30 ± 1503.37 a	632.94 ± 349.79 b	663.18 ± 211.84 b	841.42 ± 344.28 b
Hexyl acetate	1000 [27]	71.17 ± 87.43 a	33.40 ± 20.60 ab	8.85 ± 4.20 b	11.30 ± 5.05 b	9.26 ± 4.20 b
Heptyl acetate	-	0.40 ± 0.44 a	0.09 ± 0.08 b	0.00 ± 0.01 b	0.00 ± 0.00 b	0.01 ± 0.02 b
Phenethyl acetate	250 [25]	66.68 ± 45.04 a	58.00 ± 51.97 ab	32.50 ± 14.62 ab	23.61 ± 3.02 b	37.79 ± 19.14 ab
Acetic acid (mg/L)	200,000 [25]	37.31 ± 9.83 c	45.96 ± 13.08 bc	53.17 ± 12.05 bc	61.16 ± 17.52 ab	75.99 ± 26.30 a
Isobutanoic acid	20,000 [28]	2901.63 ± 794.67 abc	3506.68 ± 1230.91 ab	3807.84 ± 2048.22 a	1572.45 ± 651.55 c	2102.83 ± 560.75 bc
Hexanoic acid	420 [29]	1654.44 ± 405.83 a	1555.85 ± 192.78 a	1409.18 ± 175.31 a	1647.64 ± 359.27 a	1391.26 ± 347.36 a
Octanoic acid	500 [29]	2065.59 ± 257.39 a	1996.64 ± 181.01 ab	1789.24 ± 62.88 c	1821.23 ± 118.35 bc	1810.82 ± 141.07 bc
Decanoic acid	1000 [29]	2526.69 ± 174.48 a	2469.04 ± 98.39 a	2363.42 ± 27.35 b	2347.84 ± 40.13 b	2358.52 ± 46.73 b
Styrene	-	3.26 ± 3.22 a	0.13 ± 0.26 b	0.35 ± 0.61 b	0.60 ± 1.07 b	0.40 ± 1.13 b
Naphthalene	-	0.20 ± 0.29 b	1.78 ± 2.60 a	0.41 ± 0.34 b	0.44 ± 0.19 b	0.41 ± 0.16 b
1-Hexanol	8000 [25]	2696.95 ± 1012.55 b	2392.31 ± 909.25 b	4636.94 ± 1454.32 a	3712.82 ± 1413.24 ab	2890.52 ± 1048.65 b
(E)-3-Hexen-1-ol	1000 [30]	23.28 ± 21.82 c	46.41 ± 13.36 bc	82.05 ± 33.15 a	70.21 ± 25.95 ab	54.18 ± 37.32 abc
(E)-2-Hexen-1-ol	-	2.47 ± 2.36 b	13.66 ± 4.69 a	7.91 ± 4.18 ab	14.18 ± 5.97 a	9.34 ± 8.82 a
Nonanal	15 [31]	3.16 ± 0.91 ab	3.73 ± 1.28 a	2.63 ± 0.53 b	2.86 ± 0.82 ab	3.06 ± 0.84 ab
Ethyl lactate	100,000 [27]	79,455.87 ± 52,034.26 b	144,101.07 ± 112,565.14 ab	147,259.23 ± 59,523.25 ab	161,621.62 ± 69,605.82 ab	178,206.48 ± 96,658.57 a
Ethyl 9-decenoate	100 [32]	60.60 ± 10.18 b	108.33 ± 57.16 a	73.59 ± 12.51 b	71.48 ± 10.31 b	68.47 ± 11.99 b
Ethyl 3-methylbutanoate	3 [25]	53.68 ± 101.47 b	60.06 ± 55.18 b	180.82 ± 81.76 ab	205.26 ± 183.23 ab	299.00 ± 197.91 a
Ethyl 7-octenoate	-	14.41 ± 0.61 c	17.06 ± 2.28 a	14.96 ± 0.68 bc	14.98 ± 0.78 bc	15.74 ± 1.15 b
Ethyl 2-furoate	-	207.82 ± 166.39 b	164.16 ± 55.48 b	561.80 ± 164.48 a	663.56 ± 463.01 a	533.49 ± 348.75 a
Diethyl succinate	100,000 [33]	4388.69 ± 4118.95 b	7575.93 ± 5683.54 ab	9159.18 ± 7346.40 ab	9219.71 ± 6041.19 ab	16,445.07 ± 12,604.95 a
Ethyl phenylacetate	650 [34]	4.86 ± 3.43 a	3.53 ± 1.88 a	7.84 ± 4.95 a	7.10 ± 5.55 a	7.42 ± 5.20 a
Ethyl butanoate	20 [25]	1258.88 ± 272.63 b	1696.14 ± 319.91 ab	1545.42 ± 319.48 ab	1999.73 ± 507.79 a	1835.93 ± 1002.26 ab
Ethyl hexanoate	5 [25]	788.79 ± 231.88 a	795.16 ± 107.19 a	702.95 ± 127.01 a	861.11 ± 221.67 a	756.81 ± 282.59 a
Ethyl heptanoate	-	2.04 ± 0.97 a	1.74 ± 0.16 a	2.33 ± 0.94 a	1.81 ± 0.27 a	1.90 ± 0.29 a
Ethyl octanoate	580 [35]	2466.88 ± 424.25 a	2472.94 ± 218.44 a	2105.96 ± 127.39 a	2210.65 ± 237.12 a	2303.16 ± 448.24 a
Ethyl nonanoate	-	2.50 ± 0.73 a	2.56 ± 0.23 a	2.11 ± 0.24 a	0.59 ± 1.01 b	1.75 ± 1.48 a
Ethyl decanoate	200 [29]	1481.79 ± 683.54 a	1445.06 ± 285.06 a	804.51 ± 165.77 b	706.14 ± 215.70 b	906.88 ± 404.60 b
Ethyl dodecanoate	1500 [32]	92.08 ± 25.05 a	111.85 ± 48.88 a	57.43 ± 8.96 b	54.43 ± 10.64 b	62.44 ± 22.28 b
Ethyl tetradecanoate	-	44.49 ± 1.74 b	47.35 ± 2.86 a	42.81 ± 1.05 b	42.57 ± 1.59 b	44.43 ± 2.27 b
Ethyl hexadecanoate	-	50.08 ± 3.24 bc	55.41 ± 4.70 a	48.94 ± 2.32 c	46.97 ± 1.64 c	53.70 ± 4.87 ab
Furfural	14,100 [29]	380.26 ± 399.68 a	208.00 ± 185.12 a	477.23 ± 415.17 a	425.47 ± 514.34 a	411.58 ± 545.35 a
5-Methylfurfural	-	21.57 ± 26.66 a	9.25 ± 4.05 a	63.57 ± 67.52 a	24.35 ± 33.22 a	49.07 ± 102.85 a
Butyrolactone	100,000 [27]	7867.33 ± 2764.38 c	7670.68 ± 2714.28 c	14,149.21 ± 6046.98 ab	8700.18 ± 2291.87 bc	15,995.18 ± 7491.72 a
2-Furanmethanol	15,000 [36]	6.70 ± 6.80 b	32.11 ± 19.72 ab	132.67 ± 170.21 a	65.29 ± 62.63 ab	20.87 ± 38.25 ab
Isobutanol	40,000 [25]	65,080.39 ± 11,267.73 a	57,566.81 ± 11,746.28 a	93,732.62 ± 76,667.05 a	72,284.92 ± 32,943.52 a	70,926.97 ± 30,974.16 a
Isopentanol	65,000 [27]	155,154.36 ± 29,340.28 ab	127,002.67 ± 16,205.86 b	186,864.22 ± 48,011.05 ab	172,232.42 ± 43,205.62 ab	208,170.03 ± 82,626.75 a
4-Methyl-1-pentanol	50,000 [29]	8.94 ± 2.24 ab	5.91 ± 1.34 b	10.09 ± 2.40 a	8.69 ± 2.80 ab	12.28 ± 4.72 a
3-Methyl-1-pentanol	500 [32]	86.99 ± 34.75 ab	39.98 ± 26.43 b	104.69 ± 53.17 ab	86.93 ± 70.03 ab	128.39 ± 73.99 a
1-Octen-3-ol	20 [37]	9.55 ± 3.11 b	18.28 ± 15.21 a	17.45 ± 6.84 a	10.91 ± 2.75 ab	12.64 ± 3.51 ab
1-Heptanol	200 [35]	24.19 ± 9.23 b	19.87 ± 1.94 b	46.40 ± 17.53 a	36.41 ± 5.39 a	22.49 ± 4.01 b
*meso*-2,3-Butanediol (mg/L)	150,000 [36]	35.18 ± 25.13 a	22.42 ± 14.13 ab	35.47 ± 30.17 a	0.25 ± 0.67 b	1.39 ± 1.98 b
1-Octanol	800 [36]	24.42 ± 7.02 b	32.06 ± 7.72 a	24.01 ± 5.12 b	18.72 ± 2.87 b	19.64 ± 6.45 b
1-Decanol	400 [29]	15.03 ± 3.00 a	14.77 ± 2.39 a	12.78 ± 1.67 ab	11.27 ± 1.41 b	12.12 ± 1.61 b
Benzyl alcohol	900,000 [36]	382.85 ± 113.74 b	1258.33 ± 1172.10 a	527.11 ± 254.85 b	544.30 ± 265.40 b	436.11 ± 172.33 b
2-Phenylethanol	10,000 [25]	22,642.31 ± 7850.73 a	13,017.55 ± 2461.30 a	28,433.94 ± 14,819.99 a	17,845.87 ± 7045.54 a	30,107.99 ± 23,591.23 a
1-Dodecanol	-	2.84 ± 0.74 a	2.73 ± 0.56 ab	1.81 ± 0.73 bc	0.86 ± 1.09 cd	0.57 ± 1.06 d
(E)-β-damascenone	0.05 [25]	15.31 ± 5.79 a	16.59 ± 4.27 a	10.08 ± 2.68 b	10.41 ± 2.82 b	9.47 ± 1.70 b
Methyl octanoate	200 [38]	9.86 ± 4.62 ab	11.87 ± 1.94 a	5.86 ± 2.06 b	8.17 ± 3.68 ab	7.63 ± 4.15 ab
Propyl octanoate	-	8.80 ± 6.61 abc	13.21 ± 0.22 a	11.56 ± 4.06 ab	5.61 ± 7.00 bc	3.27 ± 6.05 c
Isobutyl octanoate	-	2.82 ± 0.81 a	2.89 ± 0.29 a	2.20 ± 0.73 ab	1.81 ± 0.82 b	1.62 ± 1.06 b
Isoamyl lactate	-	144.81 ± 114.54 a	223.67 ± 144.04 a	273.75 ± 126.53 a	275.98 ± 87.64 a	333.73 ± 251.57 a
Methyl decanoate	-	73.20 ± 16.38 a	73.76 ± 7.39 a	59.18 ± 5.67 ab	50.78 ± 22.96 b	62.68 ± 8.81 ab
Isoamyl octanoate	125 [29]	19.78 ± 10.96 a	18.66 ± 3.70 a	9.47 ± 2.55 b	8.34 ± 2.78 b	10.19 ± 5.23 b
Methyl salicylate	-	19.13 ± 4.49 b	140.69 ± 232.51 a	29.85 ± 14.90 b	19.66 ± 4.49 b	14.78 ± 7.84 b
Ethyl isopentyl succinate	-	169.46 ± 136.97 a	174.63 ± 93.52 a	232.41 ± 146.88 a	190.51 ± 76.28 a	329.11 ± 194.12 a
Methionol	500 [25]	1697.09 ± 716.09 a	907.62 ± 99.15 a	1330.56 ± 825.61 a	952.80 ± 449.08 a	1157.88 ± 923.19 a
Linalool	15 [25]	2.58 ± 0.15 c	4.70 ± 2.47 a	3.63 ± 0.37 b	0.00 ± 0.00 d	3.20 ± 0.89 bc
4-Terpineol	5000 [30]	1.03 ± 0.29 b	2.20 ± 2.36 a	1.57 ± 0.54 ab	1.12 ± 0.80 b	0.99 ± 0.66 b
β-Citronellol	100 [25]	11.72 ± 4.79 a	12.15 ± 6.29 a	6.68 ± 3.76 b	6.26 ± 5.34 b	6.81 ± 2.10 b
Geraniol	30 [25]	24.83 ± 10.43 a	33.92 ± 10.10 a	22.74 ± 2.82 ab	12.03 ± 15.29 bc	8.95 ± 12.42 c
Phenol	-	43.74 ± 25.48 a	43.35 ± 16.63 a	36.00 ± 5.01 ab	42.48 ± 9.70 a	20.49 ± 3.90 b
4-Ethylphenol	440 [31]	70.12 ± 4.50 b	66.12 ± 1.13 b	59.02 ± 20.75 b	182.44 ± 115.00 a	49.09 ± 89.82 b

* Averages and standard deviation followed by different letters are significant at *p* < 0.05, Duncan’s multiple range test. ^†^ The odor threshold values were determined in 10% *w*/*w* water/ethanol solution [22,32,33]; beer [23]; 14% *v*/*v* water/ethanol solution [24,25,27]; 11% *v*/*v* water/ethanol with 7 g/L of glycerin and 5 g/L of tartaric acid, pH 3.4 [26]; 10% *v*/*v* water/ethanol solution with 5 g/L of tartaric acid, pH 3.2 [28]; 9.72 g/100 g ethanol/water mixture with 5 g/L of tartaric acid, pH 3.2 [29]; water [30]; 12% *v*/*v* ethanol/water with 5 g/L of tartaric acid, pH 3.2 [31]; 12% *v*/*v* ethanol/water, 5 g/L tartaric acid, pH 3.5 [34,35].

**Table 3 foods-11-00787-t003:** CIELab parameters (*L*, *a*, *b*, *H*_ab_, and *C*_ab_) in Marselan wines from different regions in China *.

CIELab	Jiaodong Peninsula	Bohai Bay	Huaizhuo Basin	Loess Plateau	Xinjiang
*L*	66.32 ± 13.31 a	61.54 ± 20.24 ab	42.58 ± 10.02 c	48.90 ± 8.62 bc	39.11 ± 12.07 c
*a*	34.74 ± 11.95 b	34.49 ± 14.90 b	49.98 ± 8.76 a	45.64 ± 9.33 ab	52.52 ± 12.07 a
*b*	6.94 ± 5.50 a	7.10 ± 1.58 a	7.91 ± 3.79 a	9.22 ± 4.43 a	9.07 ± 7.95 a
*H* _ab_	10.78 ± 5.62 a	13.70 ± 8.28 a	9.76 ± 6.27 a	11.46 ± 5.12 a	11.18 ± 8.74 a
*C* _ab_	35.62 ± 12.56 b	35.39 ± 14.41 b	50.85 ± 8.02 a	46.72 ± 9.43 ab	53.93 ± 11.42 a

* Averages and standard deviation followed by different letters are significant at *p* < 0.05, Duncan’s multiple range test.

**Table 4 foods-11-00787-t004:** Concentrations of the phenolic compounds of Marselan wines from different regions of China *.

Compound	Jiaodong Peninsula	Bohai Bay	Huaizhuo Basin	Loess Plateau	Xinjiang
Myricetin-3-*O*-galactoside	0.04 ± 0.09 a	0.08 ± 0.19 a	0.05 ± 0.11 a	0.07 ± 0.09 a	0.17 ± 0.30 a
Myricetin-3-*O*-glucoside	2.38 ± 1.31 b	1.97 ± 4.19 b	2.71 ± 2.73 b	1.62 ± 1.59 b	14.68 ± 3.83 a
Dihydroquercetin	1.40 ± 0.66 a	0.95 ± 0.50 ab	1.00 ± 0.45 ab	0.82 ± 0.50 ab	0.63 ± 0.38 b
Quercetin-3-*O*-glucuronide	0.53 ± 0.46 c	1.40 ± 2.45 bc	1.68 ± 1.44 bc	3.04 ± 2.36 ab	5.10 ± 2.26 a
Quercetin-3-*O*-galactoside	0.20 ± 0.19 b	0.27 ± 0.57 b	0.29 ± 0.32 b	0.38 ± 0.28 ab	0.84 ± 0.61 a
Quercetin-3-*O*-glucoside	0.00 ± 0.00 b	0.02 ± 0.04 b	0.08 ± 0.26 b	0.00 ± 0.00 b	4.66 ± 4.13 a
Dihydrokeampferol	0.44 ± 0.10 a	0.52 ± 0.24 a	0.63 ± 0.21 a	0.51 ± 0.26 a	0.53 ± 0.28 a
Syringetin-3-*O*-glucoside	2.65 ± 0.62 b	3.36 ± 0.76 b	2.72 ± 0.94 b	4.69 ± 1.22 a	4.39 ± 0.89 a
Isorhamnetin-3-*O*-glucoside	0.08 ± 0.13 b	0.19 ± 0.48 b	0.14 ± 0.34 b	0.07 ± 0.19 b	1.58 ± 1.04 a
Procyanin B1	32.91 ± 9.91 ab	36.35 ± 27.27 ab	35.66 ± 8.64 ab	25.87 ± 19.83 b	44.06 ± 15.36 a
Gallocatechin	4.26 ± 2.02 ab	2.25 ± 0.54 c	2.85 ± 1.06 bc	2.36 ± 1.34 c	4.36 ± 0.55 a
Catechin	24.57 ± 8.43 a	28.26 ± 14.46 a	20.83 ± 4.14 a	19.29 ± 10.50 a	28.05 ± 12.67 a
Procyanin C1	5.38 ± 3.55 ab	6.50 ± 6.57 a	4.88 ± 2.36 ab	2.14 ± 3.50 b	8.13 ± 6.91 a
Procyanin B2	12.71 ± 5.04 ab	15.62 ± 11.53 a	13.49 ± 3.88 ab	8.98 ± 7.70 b	18.31 ± 7.84 a
Epicatechin	39.99 ± 16.27 ab	43.69 ± 23.87 a	30.09 ± 8.80 ab	25.55 ± 17.65 b	42.53 ± 17.50 ab
Gallic acid	19.95 ± 5.87 bc	12.95 ± 9.55 c	31.93 ± 9.98 a	25.30 ± 15.63 ab	25.30 ± 5.92 ab
Protocatechuic acid	1.38 ± 0.57 a	1.61 ± 1.11 a	2.69 ± 3.41 a	2.17 ± 1.35 a	0.89 ± 0.36 a
2-Hydroxybenzoic acid	0.12 ± 0.10 ab	0.08 ± 0.16 ab	0.24 ± 0.29 a	0.12 ± 0.13 ab	0.00 ± 0.00 b
Caffeic acid	10.00 ± 5.14 a	12.06 ± 9.27 a	10.59 ± 10.10 a	8.58 ± 8.30 a	6.38 ± 8.37 a
4-Hydroxycinnamic acid	4.85 ± 2.64 a	5.57 ± 1.90 a	3.83 ± 3.04 a	4.31 ± 3.12 a	3.00 ± 2.69 a
Ferulic acid	0.05 ± 0.06 a	0.21 ± 0.18 a	0.10 ± 0.17 a	0.03 ± 0.07 a	0.09 ± 0.12 a
Delphinidin-3-*O*-glucoside	6.56 ± 2.97 ab	3.91 ± 1.26 b	10.94 ± 7.24 a	3.78 ± 2.59 b	6.75 ± 3.13 ab
Cyanidin-3-*O*-glucoside	0.44 ± 0.14 b	0.39 ± 0.10 b	1.24 ± 0.96 a	0.40 ± 0.25 b	0.43 ± 0.13 b
Petunidin-3-*O*-glucoside	4.66 ± 1.88 ab	3.07 ± 0.96 b	6.33 ± 4.09 a	3.12 ± 2.11 b	4.33 ± 2.51 ab
Peonidin-3-*O*-glucoside	2.47 ± 0.79 ab	1.73 ± 0.69 b	3.73 ± 2.62 a	1.74 ± 1.12 b	1.95 ± 0.99 ab
Malvidin-3-*O*-glucoside	156.24 ± 43.75 a	110.71 ± 45.35 a	111.27 ± 58.38 a	120.86 ± 105.00 a	99.42 ± 51.61 a
Delphinidin-3-*O*-acetylglucoside	2.09 ± 1.39 ab	1.60 ± 0.33 b	2.71 ± 1.37 a	1.35 ± 0.53 b	2.06 ± 1.17 ab
Petunidin-3-*O*-acetylglucoside	1.96 ± 1.70 a	1.14 ± 0.43 a	2.04 ± 1.45 a	1.03 ± 0.89 a	1.60 ± 1.16 a
Peonidin-3-*O*-acetylglucoside	1.59 ± 0.75 a	1.34 ± 0.84 a	1.49 ± 1.12 a	1.34 ± 1.29 a	1.03 ± 0.72 a
Malvidin-3-*O*-acetylglucoside	78.57 ± 34.30 a	54.57 ± 26.21 a	43.45 ± 26.85 a	43.10 ± 43.35 a	46.43 ± 30.73 a
Delphinidin-3-*O*-coumaroylglucoside (*cis + trans*)	0.85 ± 0.14 a	0.71 ± 0.09 a	0.91 ± 0.27 a	0.70 ± 0.14 a	0.95 ± 0.37 a
Petunidin-3-*O*-coumaroylglucoside *(cis + trans)*	0.49 ± 0.27 a	0.20 ± 0.17 a	0.38 ± 0.37 a	0.21 ± 0.24 a	0.25 ± 0.22 a
Peonidin-3-*O*-coumaroylglucoside *(cis + trans)*	1.23 ± 0.57 a	0.83 ± 0.66 a	0.84 ± 0.77 a	0.81 ± 0.93 a	0.52 ± 0.50 a
Malvidin-3-*O*-coumaroylglucoside *(cis + trans)*	32.38 ± 14.09 a	18.51 ± 14.84 ab	13.86 ± 11.34 b	17.04 ± 20.98 ab	11.96 ± 9.07 b
Vinylformic acid adduct of Malvidin-3-*O*-glucoside (Vitisin A)	11.26 ± 8.00 a	12.44 ± 9.31 a	16.81 ± 6.19 a	15.68 ± 4.54 a	14.66 ± 8.01 a
Vinylformic acid adduct of Malvidin-3-*O*-acetylglucoside	7.88 ± 10.34 a	6.05 ± 4.36 a	7.07 ± 3.75 a	5.64 ± 2.79 a	6.34 ± 3.18 a
Vinylformic acid adduct of Malvidin-3-*O*-coumaroylglucoside	2.82 ± 1.39 b	5.12 ± 6.09 a	3.10 ± 1.00 ab	2.70 ± 1.00 b	3.57 ± 1.61 ab
Acetaldehyde adduct of Malvidin-3-*O*-glucoside (Vitisin B)	24.11 ± 21.10 a	9.16 ± 4.63 a	32.86 ± 43.65 a	39.29 ± 51.29 a	23.01 ± 19.80 a
Acetaldehyde adduct of Malvidin-3-*O*-acetylglucoside	10.64 ± 9.17 a	4.58 ± 2.11 a	11.85 ± 16.27 a	15.77 ± 22.79 a	9.62 ± 6.73 a
Acetaldehyde adduct of Malvidin-3-*O*-coumaroylglucoside	7.74 ± 7.27 a	2.26 ± 0.49 a	7.68 ± 11.05 a	7.11 ± 9.55 a	5.60 ± 6.11 a
Vinylphenol adduct of Malvidin-3-*O*-glucoside	22.23 ± 18.87 a	23.63 ± 12.48 a	15.60 ± 11.72 a	21.14 ± 8.83 a	11.61 ± 10.23 a

* Averages and standard deviation followed by different letters are significant at *p* < 0.05, Duncan’s multiple range test.

**Table 5 foods-11-00787-t005:** Sensory profiles of Marselan wines from different regions of China *.

Attributes	Jiaodong Peninsula	Bohai Bay	Huaizhuo Basin	Loess Plateau	Xinjiang
Hue	9.58 ± 0.04 a	9.3 ± 0.29 a	9.14 ± 0.57 a	9.83 ± 0.06 a	9.41 ± 0.18 a
Color intensity	7.93 ± 0.20 c	8.91 ± 0.76 ab	8.42 ± 0.10 bc	9.44 ± 0.03 a	9.21 ± 0.23 a
Floral	6.57 ± 0.66 a	6.51 ± 0.86 a	4.26 ± 0.67 b	5.87 ± 0.23 a	5.62 ± 0.54 a
Fruity	7.9 ± 0.02 a	7.23 ± 0.47 a	6.26 ± 0.34 b	7.24 ± 0.37 a	7.50 ± 0.44 a
Hebaceous	5.23 ± 0.84 a	4.56 ± 0.24 a	3.82 ± 1.03 a	4.14 ± 0.33 a	4.3 ± 0.89 a
Acidiy	7.44 ± 0.53 a	7.21 ± 0.83 ab	6.41 ± 0.11 ab	7.28 ± 0.04 ab	5.99 ± 0.88 b
Astringency	5.32 ± 0.56 a	5.44 ± 0.59 a	5.85 ± 0.62 a	6.39 ± 0.38 a	6.46 ± 0.32 a

* Averages and standard deviation followed by different letters are significant at *p* < 0.05, Duncan’s multiple range test.

## Data Availability

All data generated or analyzed during this study are included in this article.

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

*vinifera* cv. Cabernet Sauvignon wines from five wine-growing regions in China. Food Chem..

