# Peer review of "Regional Variation of Chemical Characteristics in Young Marselan (*Vitis vinifera* L.) Red Wines from Five Regions of China"

_foods, 2022, doi:10.3390/foods11060787_

Round 1

Reviewer 1 Report

This manuscript “ Regional Variation of Chemical Characteristics in Young  Marselan (Vitis vinifera L.) Red Wines from Five Regions of China “ aims to investigate the chromatographic fingerprints of Marselan wines from different wine-producing regions in China and identify the flavor compound characteristics in different regions.

This is a well-written paper.

The introduction cites previous works, the methodology used was the appropriate one and the data obtained is consistent and important, once it helps to distinguish the wines according to their region.

Maybe the authors should enlarge the figures a little bit more so that they will be easier to read.

So, in my opinion, the article only needs a minor revision.

Author Response

Thanks for your suggestions and comments that helped to improve the quality of the manuscript.

Comment: Maybe the authors should enlarge the figures a little bit more so that they will be easier to read.

Response: Figure 1 and figure 2 have been improved in quality and updated in the revised manuscript.

Reviewer 2 Report

Reviewer comment for manuscript: Regional Variation of Chemical Characteristics in Young Marselan (Vitis vinifera L.) Red Wines from Five Regions of China

Dear Authors,

  1. The title and the abstract are the most important parts of a research paper and should be pleasant to read. The purpose of the abstract is to summarize the research paper by stating the purpose, relevance and novelty degree of the research, the experimental method, the findings, and the precise conclusions. Please revise the abstract, so that the most important results are presented clearly, with less focus on the conclusions.
  2. The paper presents average values of the physico-chemical characteristics of the analyzed wines from different regions, but also from different years. Such an interpretation cannot be made because the harvest year influences those values significantly due to the specific climatic conditions. Thus, average values can be achieved on samples from the same year, but the number of samples per year is not the same, therefore it is not relevant how to present those values. The same observation applies to chromatographic results. Please review the interpretation and rendering of the usual physicochemical characteristics.
  3. Please also present in the Materials and methods section, the specific methods for determining the values of the physico-chemical parameters even if a Foss device was used to obtain them.
  4. The concentration of SO2 in the analyzed samples was not determined, an important value that influences the sensory characteristics of the wines.
  5. It is not specified in the Materials and methods section in which style the wine samples were obtained, if the same technology was used. Please clarify.
  6. Please replace the links in the text with bibliographic indications and then transfer them to the Bibliography section.
  7. Line 105 - please explain how the CIE-Lab method has been simplified. Please consult the compendium of OIV methods and bring appropriate bibliographic arguments and indications.
  8. Line 156 - please briefly present the methods you are talking about even if you have indicated the bibliographic source.
  9. Line 195 - please present bibliographic information and a brief presentation of the Köppen-Geiger climate classification system.
  10. We recommend MAJOR Revisions with complete presentation of the physico-chemical and chromatographic results for each analyzed sample and the rethinking of their interpretation per harvest year.

Reviewer 3 Report

The manuscript titled “Regional Variation of Chemical Characteristics in Young Marselan (Vitis vinifera L.) Red Wines from Five Regions of China” related to the evaluation of aroma profile and phenolics profile of Marselan wines obtained from 5 Chinese regions. Experimental setup same as an idea are appropriate since the quality of monovarietal wines differs between wine regions due to the complexity of climate, agricultural, and winemaking practices. The manuscript is generally well written and well organized. Other studies related to Marselan wines were duly cited throughout the entire manuscript. Applied analytical methods are appropriate. The conclusions are supported with results that are presented in a clear manner. There are several minor issues that should be addressed.

Title and materials and methods – Please add information when wines were analysed, because if wines were analysed in 2021 or similar, these can’t be considered as young wines.

Section 3.1. If any exits, please add literature data of optimal climate conditions for the cultivation of Marselan grape variety and compare it with climate data obtained in Chinese regions. Comparison with French regions in which Marselan is cultivated would be appreciated.

Table 2 Please add odor threshold for all identified aroma compounds and provide references. Also please add some discussion about aroma descriptors (like for geraniol Lines 262-263) for all compounds relevant for Marselan variety, and correlate this information with sensory attributes.

Author Response

We would like to appreciate your comments and suggestions which helped us to improve the quality of the manuscript.

Comment 1: Title and materials and methods – Please add information when wines were analysed, because if wines were analysed in 2021 or similar, these can’t be considered as young wines.

Response: Thanks for your comments. All the analyses were completed at the end of 2017. The detail descriptions were included in the section of “2.2. Wine Samples”. That’s why we pointed out the young wines were used in this study (Line 90-92).

Comment 2: Section 3.1. If any exits, please add literature data of optimal climate conditions for the cultivation of Marselan grape variety and compare it with climate data obtained in Chinese regions. Comparison with French regions in which Marselan is cultivated would be appreciated.

Response: Thanks for your comments. The climate of a specific wine region could make the wine with characteristic style. As we know, Marselan is planted in Languedoc and southern Rhone of France with largest area in the world compared to other countries. Additionally, Marselan has been approved for use in Bordueaux and the first planting began in 2021. Except for China, Marselan is also planted in several countries, such as Brazil, Spain and Argentina, but with small planting area. Although the planting area of Marselan in China is growing, but we cannot define the optical climate conditions for Marselan. In conclusion, Marselan is still a new grape variety for us, athough it has been bred for many years.

         Just like your comments, we would like to compare the differences in chemicals and sensory profiles among Marselan wines from different regions. We researched the literatures related to Marselan and approximately 45 literatures were collected. Unfortunately, we could not find any studies related to the flavor chemistry of Marselan in France, as well as in other countries. Therefore, further studies should be carried out to improve the knowledge of Marselan, which will be helpful to improve the Marselan wine quality through viticultural and enological practices.

Comment 3: Table 2 Please add odor threshold for all identified aroma compounds and provide references. Also please add some discussion about aroma descriptors (like for geraniol Lines 262-263) for all compounds relevant for Marselan variety, and correlate this information with sensory attributes.

Response: Thanks for your comments. The odor thresholds for volatile compounds were added in the Table 2 (Line 257-289). Some discussion about odor threshold and aroma descriptor were added, which can be found in the revised manuscript, such as Line 340-344 and Line 411-418.